# UNIVERSALITY OF DEEP NEURAL NETWORK LOTTERY TICKETS: A RENORMALIZATION GROUP PERSPECTIVE

## ABSTRACT

Foundational work on the Lottery Ticket Hypothesis has suggested an exciting corollary: winning tickets found in the context of one task can be transferred to similar tasks, possibly even across different architectures. While this has become of broad practical and theoretical interest, to date, there exists no detailed understanding of why winning ticket universality exists, or any way of knowing whether tickets can be transferred between two given tasks, without first performing transfer experiments. To address these outstanding open questions, we make use of renormalization group theory, one of the most successful tools in theoretical physics. We find that iterative magnitude pruning, the method used for discovering winning tickets, is a renormalization group scheme. This opens the door to a wealth of existing numerical and theoretical tools, some of which we leverage here to examine winning ticket universality in large scale lottery ticket experiments, as well as sheds new light on the success iterative magnitude pruning has found generally in machine learning.

## 1 INTRODUCTION

The lottery ticket hypothesis (LTH) for deep neural networks (DNNs) proposes that DNNs contain sparse subnetworks that can be trained in isolation and can reach performance that is equal to, or better than, that of the full DNN in the same number of training iterations (Frankle & Carbin, 2019; Frankle et al., 2020a). These subnetworks are called winning lottery tickets. The LTH has provided paradigm shifting insight into the success of dense DNNs, suggesting a key role for the emergence of winning tickets with increasing size, reminiscent of the maxim "more is different" (Anderson, 1972). In recent years, researchers have found an intriguing corollary: winning tickets found in the context of one task can be transferred to related tasks (Desai et al., 2019; Mehta, 2019; Morcos et al., 2019; Soelen & Sheppard, 2019; Chen et al., 2020a; Gohil et al., 2020; Chen et al., 2021a; Sabatelli et al., 2021), possibly even across different architectures (Chen et al., 2021c). In addition to having applications of practical interest, these results imply that winning tickets can be used to study how tasks and architectures are "similar". However, to date, there exists no principled understanding of why winning tickets can be transferred between tasks, nor does there exist any way to know, without directly performing transfer experiments, which previously studied tasks a given winning ticket can be transferred to. Furthermore, there is a general lack of theoretical work on iterative magnitude pruning (IMP) (Elesedy et al., 2021), the most commonly used method to find winning tickets.

This is in striking analogy to the state of statistical physics in the early-to-mid–20[th] century. Empirical evidence suggested that disparate systems, governed by seemingly different underlying physics, exhibited the *same*, universal properties near their phase transitions. While heuristic methods provided insight (Kadanoff, 1966), a full theory from first principle on this universality was not realized until the development of the renormalization group (RG) (Wilson, 1971a;b; 1975).

RG theory has not only provided a framework for explaining universal behavior near phase transitions, but also a scheme for grouping systems by that behavior. This classification, introduced via the notion of universality classes, has allowed for a detailed understanding of materials (Tougaard, 1997; Winter & Mours, 1997; Durin & Zapperi, 2000; Bonilla et al., 2010), and provides immediate knowledge as to what other classified systems a given material behaves like.

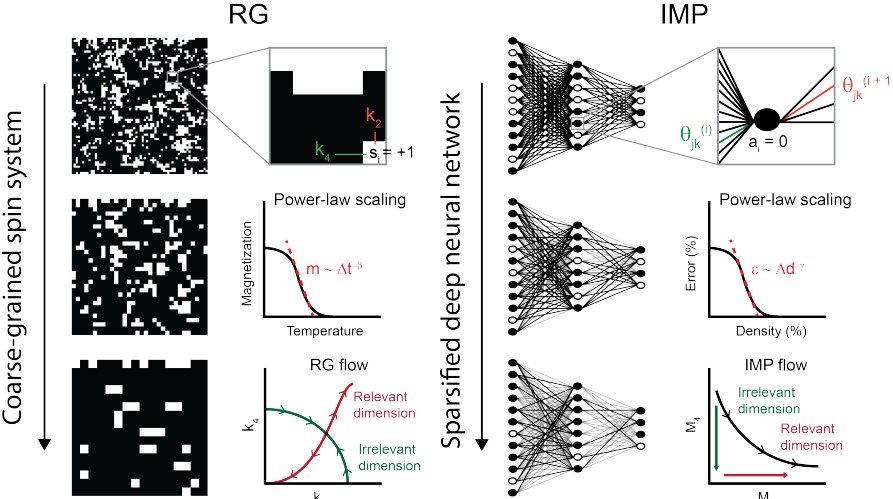

Figure 1: **Summary of the similarities between the RG and IMP.** Both the RG and IMP are applied iteratively to coarse-grain systems, revealing "relevant" features. Certain observables are known to have regimes where they follow power-law scaling. In the case of the RG, this scaling and its universality are associated with properties of the flow the RG induces in the space of coupling constants. The nature of the flow IMP induces has not been previously studied, but none-the-less exists. See Table 1 for the analogous quantities in each theory.

The analogy between universality in RG and LTH theories becomes strengthened under the following consideration. Recently, Rosenfeld et al. (2020) found that when the density (i.e. the percentage of parameters remaining) of a DNN being pruned via IMP is in a certain range, $d_L < d < d_C$, the network's error scales according to the following power-law,

$$e \sim (d_C - d)^{-\gamma} = \Delta d^{-\gamma}. \tag{1}$$

Power-law scaling is well known to emerge in critical phenomena. For instance, when the temperature of a classical spin system (e.g. two-dimensional Ising model) is near and below the critical temperature, $t < t_C$, many observables (e.g. magnetization) exhibit power-law scaling,

$$m \sim (t_C - t)^{-\beta} = \Delta t^{-\beta}. \tag{2}$$

These similarities hint at a more fundamental connection between the universality present in winning tickets and in physical systems near phase transitions (see Fig. 1 and Table 1). Indeed, we show that *IMP is an RG scheme* (i.e. IMP satisfies the properties required to be considered an RG operator). By analyzing winning ticket transfer experiments (Chen et al., 2021a), we find that tasks that allow for successful transfer of winning tickets have similar behavior in their IMP flow (i.e. the way in which the parameters of the DNN change after each iteration of IMP), supporting a key prediction of the theory. We additionally leverage the RG framework to interpret recent results on transferring winning tickets across differing architectures (Chen et al., 2021c). Our contributions are the following:

- A theoretical basis for understanding IMP and the universality of winning tickets;

- Experimental support of the theory in large scale lottery ticket experiments, providing insight into previously known heuristic based results;

- A new set of tools for studying the LTH and associated phenomena.

## 2 THE RENORMALIZATION GROUP AND ITERATIVE MAGNITUDE PRUNING

The RG operator, $\mathcal{R}$, can be viewed as a method for "coarse-graining". That is, each application of $\mathcal{R}$ replaces local degrees of freedom with a composite of their values[1]. An example of this for the two-dimensional Ising model is shown in the left-hand column Fig. 1, where neighborhoods of four spins are replaced by their mode.

The formal way to study $\mathcal{R}$ is to consider its action on a given Hamiltonian (i.e. energy function). For classical spin systems (e.g. two-dimensional Ising model), $\mathcal{H}$ has the general form

$$\mathcal{H}(\mathbf{s}, \mathbf{k}) = -\sum_i k_1 s_i - \sum_{\langle i,j \rangle} k_2 s_i s_j - ..., \tag{3}$$

where the $s_i$ are the spins of the system (e.g. $s_i \in \{-1, +1\}$), $\langle \cdot, \cdot \rangle$ represents sites on the lattice that are nearest neighbors, and the $k_i$ are the strengths of the different coupling constants (e.g. $k_2$ is the strength of the nearest neighbor coupling).

Due to the fact that coarse-graining amalgamates spins, the spin system resulting from an application of $\mathcal{R}$ can be viewed as equivalent to the original, but with a new set of coupling constants. Therefore, the coarse-grained system has a different Hamiltonian, which is given by

$$\mathcal{R}\mathcal{H}(\mathbf{s}, \mathbf{k}) = \mathcal{H}(\mathbf{s}', \mathcal{T}\mathbf{k}) = \mathcal{H}(\mathbf{s}', \mathbf{k}'), \tag{4}$$

where $\mathbf{s}'$ is the new set of spins and $\mathbf{k}'$ is the new set of couplings determined by the operator $\mathcal{T} : \mathbb{R}^K \to \mathbb{R}^K$. Here $K$ is the maximum number of couplings considered, usually introduced to keep the considered operators finite.

$\mathcal{R}$ can be applied iteratively, defining a flow through the function space of Hamiltonians via Eq. 4, with an associated flow in the space of coupling constants. The latter is commonly referred to as the RG flow. While $\mathcal{R}$ is often a complicated, non-linear function, $\mathcal{T}$ can be linearized near fixed points (Goldenfeld, 2005). The flow will grow in the direction of the eigenvectors $\mathbf{v}_r$ with eigenvalues $\lambda_r > 1$ and will shrink along the direction of the eigenvectors $\mathbf{v}_i$ with $\lambda_i < 1$. These eigenvectors, which are linear combinations of coupling constants are called **relevant** and **irrelevant**, respectively. A schematic of these, and the general RG flow, is presented in Fig. 1. By examining the effect of applying $\mathcal{R}$ to a given system, and its flow in the space of coupling constants, it is possible to find which components of the system are necessary for certain macroscopic behavior (i.e those that are relevant), and which are not. Systems belonging to a particular universality class have the same relevant directions.

Interestingly, many of the properties discussed above can be analogously found for IMP. To see this, consider a DNN with loss function $\mathcal{L}(\mathbf{a}, \boldsymbol{\theta})$, where $\mathbf{a}$ is the unit activations and $\boldsymbol{\theta}$ is the DNN parameters. Let $\mathcal{I}$ represent a single application of the IMP process. The DNN pruned via an application of $\mathcal{I}$ is given by the relation

$$\mathcal{I}\mathcal{L}(\mathbf{a}, \boldsymbol{\theta}) = \mathcal{L}(\mathbf{a}', \mathcal{T}\boldsymbol{\theta}) = \mathcal{L}(\mathbf{a}', \boldsymbol{\theta}'), \tag{5}$$

where the new set of parameters, $\boldsymbol{\theta}'$, are given by an operator $\mathcal{T}$. This new set of parameters leads to a new, coarse-grained set of activations, $\mathbf{a}'$. The similarities between Eqs. 4 and 5 are striking.

We note here that, in the case of IMP, $\mathcal{T}$ is the composition of a masking operator, $\mathcal{M}$, and a refining operator $\mathcal{F}$. That is, $\mathcal{T} = \mathcal{F} \circ \mathcal{M} : \mathbb{R}^N \to \mathbb{R}^N$, where $N$ is the number of parameters of the

---

[1]Note that here we consider the block spin, or Kadanoff, RG. The momentum space, or Wilsonian, RG has a different, but related, interpretation.

Table 1: Analogous quantities in RG and IMP theory.

| RG | IMP |
|---|---|
| Spins ($s_i$) | Unit activations ($a_i$) |
| Coupling constants ($k_i$) | Parameters ($\theta_i$) |
| Hamiltonian ($\mathcal{H}[\mathbf{s}, \mathbf{k}]$) | Loss function ($\mathcal{L}[\mathbf{a}, \boldsymbol{\theta}]$) |

DNN. $\mathcal{M}$ is defined via the pruning procedure implemented (e.g. magnitude pruning). While we are considering IMP here, because of its connection to the Lottery Ticket Hypothesis (Frankle & Carbin, 2019; Frankle et al., 2020a), there are many other pruning procedures [e.g. Hessian based pruning (LeCun et al., 1989)], all of which have their own $\mathcal{M}$. Similarly, $\mathcal{F}$ is defined via the refinement procedure used, making it dependent on the choice of optimizer and whether or not the DNN parameters are left alone or "rewound" to their value at a previous point during training (Frankle et al., 2020a; Renda et al., 2020).

For $n$ iterations of $\mathcal{I}$, the resulting network is given by

$$\mathcal{I}^n \mathcal{L}(\mathbf{a}^{(0)}, \boldsymbol{\theta}^{(0)}) = \mathcal{L}(\mathbf{a}^{(n-1)}, \mathcal{T}^n \boldsymbol{\theta}^{(0)}) = \mathcal{L}(\mathbf{a}^{(n-1)}, \boldsymbol{\theta}^{(n-1)}). \tag{6}$$

This defines a trajectory in parameter space, $\boldsymbol{\theta}^{(0)} \to \boldsymbol{\theta}^{(1)} \to ... \to \boldsymbol{\theta}^{(n-1)}$, which we will refer to as the **IMP flow**. A schematic illustration of what the IMP flow might look like is given in Fig. 1 (the axes of which will be explained in Sec. 3.1). Just as in the case of RG theory, the directions of the IMP flow are determined by the eigenvectors of $\mathcal{T}$, which grow or shrink exponentially by the magnitude of their associated eigenvalues.

## 2.1 CONNECTION BETWEEN THE RG AND STANDARD LTH FRAMEWORKS

Before we formally show that IMP is an RG scheme, we discuss how the standard theory that has emerged from studying the LTH is related to RG theory.

In the familiar picture, the success of winning tickets is attributed to the surviving sparse DNN being able to find a "good" local minimum in the loss landscape (Frankle & Carbin, 2019; Frankle et al., 2020a). This may be possible because the training of DNNs via stochastic gradient descent (SGD) appears to be rapidly confined to a low-dimensional subspace (Gur-Ari et al., 2018), implying that DNNs only "feels" changes to a small number of parameters during much of training. Parameters that are not in this low-dimensional subspace can, therefore, be removed with minimal impact. If a sparse DNN is initialized in this subspace (as late rewinding aims to do), then it may be possible for training to find the same, or related, local minima as the full DNN (Evci et al., 2020; Maene et al., 2021).

In this way, the parameters that do not lie in the low-dimensional subspace are irrelevant, and repeated applications of IMP are expected to remove them. On the other hand, the relevant parameters are those that span this subspace, and their removal changes the local minimum that the sparse DNN converges to. Therefore, certain observable functions of the DNN, such as error, are expected to only be sensitive to the relevant, but not irrelevant, parameters.

If two models share the same low-dimensional subspace, then they should be able to transfer winning tickets. Note that this transferability is highly non-trivial. Indeed, it was only with the development of the LTH, and subsequent experimental work, that this was considered to be a possibility (Frankle & Carbin, 2019; Mehta, 2019; Morcos et al., 2019). It is, in general, difficult to find these subspaces and accurately compare them across experiments. RG theory provides us with tools for finding relevant and irrelevant directions, which do not rely on explicitly constructing the underlying subspace. In particular, RG theory says that if two models have the same eigenvectors of $\mathcal{T}$ with eigenvalues > 1, then they have the same relevant parameters. This means that, once the eigenvalues are computed, we will know whether winning tickets can be transferred between two models, without having to run any additional experiments. Note that the minimal sparsity a winning ticket can have, and still successfully be transferred, is not identified.

## 2.2 IMP AS AN RG SCHEME

To make the connection between IMP and RG more precise, we show that IMP fits the definition of an RG scheme. To do this, we consider the projection operator, $\mathcal{P}$, that is associated with the RG operator. In the case of classical spin systems, the projection operator maps the spins, $s_i$, to a coarse-grained spin system, $s'_I$, such that it satisfies

$$\text{Tr}_{\{s_i\}} \mathcal{P}(s_i, s'_I) \exp[\mathcal{H}(s_i, \mathbf{k})] = \exp[\mathcal{H}(s'_I, \mathbf{k}')], \tag{7}$$

where $\text{Tr}_{\{s_i\}}$ is the trace operator over the values that the $s_i$ can take (e.g. $\pm 1$) (Goldenfeld, 2005). For the two-dimensional Ising model, it is standard to take

$$\mathcal{P}(s_i, s_I') = \prod_I \delta \left[ s_I' - \text{sign} \left( \sum_{j \in I} s_j \right) \right], \tag{8}$$

where $\delta$ is the Kronecker delta function and $\text{sign}(\cdot)$ is $+1$ if the argument is positive and $-1$ if it is negative (Goldenfeld, 2005). Eq. 8 formally defines the mapping from $s_i \to s_I'$ as

$$s_I' = \text{sign} \left( \sum_{j \in I} s_j \right). \tag{9}$$

The projection operator is not unique, but must satisfy the following three properties:

1. $\mathcal{P}(s_i, s_I') \geq 0$;
2. $\mathcal{P}(s_i, s_I')$ respects the symmetry of the system;
3. $\sum_{\{s_I'\}} \mathcal{P}(s_i, s_I') = 1$.

To find a projection operator associated with $\mathcal{I}$, we start by finding a mapping between the activations of all the units, $\mathbf{a}$, before and after an application of IMP. This is because $\mathbf{a}$, and not $\boldsymbol{\theta}$, is the analogous quantity to $\mathbf{s}$ (Table 1 and Fig. 1). Without loss of generality, we can consider the activation of unit $j$ in layer $i$ as being defined by

$$a_j^{(i)} = h \left[ \sum_k g_k(\mathbf{a}, \boldsymbol{\theta}) \right], \tag{10}$$

where $h$ is the activation function (e.g. ReLu, sigmoid) and the $g_k$ are functions that determine how the different parameters and activations of other units affect $a_j^{(i)}$. For instance, in a feedforward DNN, the impact of the bias of unit $j$ in layer $i$ is given by $g_0 = \theta_j^{(i)}$, and the weighted input from the previous layer is given by $g_1 = \sum_{k=1}^{N^{(i-1)}} \theta_{jk}^{(i)} a_k^{(i-1)}$. Here $N^{(i-1)}$ is the number of units in layer $i-1$.

As discussed above, IMP changes the parameters $\boldsymbol{\theta}$ by the operator $\mathcal{T}$, which is defined by a composition of $\mathcal{M}$ and $\mathcal{F}$. Therefore, the activation of unit $j$ in layer $i$, after applying $\mathcal{I}$, is given by

$$a_j'^{(i)} = h \left[ \sum_k g_k(\mathbf{a}, \mathcal{F} \circ \mathcal{M} \boldsymbol{\theta}) \right], \tag{11}$$

and thus, the projection operator associated with $\mathcal{I}$ is

$$\mathcal{P}\left( a_j^{(i)}, a_j'^{(i)} \right) = \prod_{j=1}^N \delta \left\{ a_j'^{(i)} - h \left[ \sum_k g_k(\mathbf{a}, \mathcal{F} \circ \mathcal{M} \boldsymbol{\theta}) \right] \right\}. \tag{12}$$

We can easily verify that the projection operator defined by Eq. 12 satisfies all three properties needed for an RG projection operator. First, property #1 is satisfied, as Eq. 12 is a product of Kronecker delta functions. Property #2 is satisfied, as IMP only removes parameters, so the form of $a_j^{(i)}$ given in Eq. 11 and, thus, the loss function, will remain intact until layer collapse (i.e. when all the weights from one layer to another are removed). Finally, for property #3 to be satisfied, we can fix the test and training sample ordering for each epoch (as is done when the random seed is fixed). In such a case, both the mask and refining operators in Eq. 12 are deterministic and $\mathcal{P}\left( a_j^{(i)}, a_j'^{(i)} \right)$ will be unique.

Having shown that these three properties are satisfied by the projection operator associated with IMP, we have found that $\mathcal{I}$ meets the criteria for being an RG operator, making it an RG scheme.

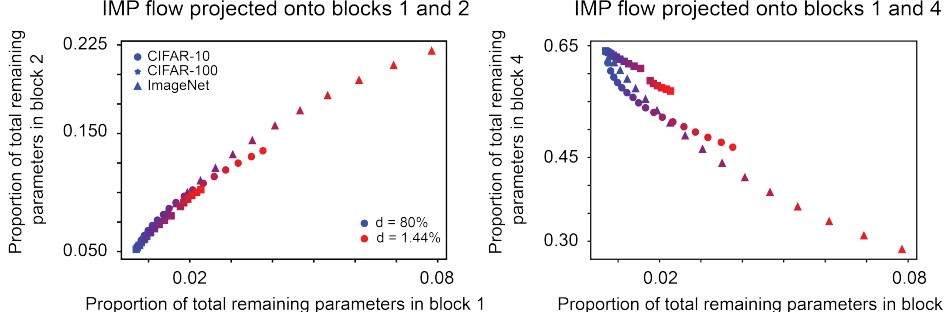

Figure 2: **IMP flow.** Two-dimensional projections of the IMP flow for ResNet-50, pre-trained on ImageNet via self-supervised simCLR training (Chen et al., 2020b), applied to three different tasks: ImageNet (triangles), CIFAR-10 (circles), and CIFAR-100 (stars). Blue corresponds to a single application of IMP (density equal to $80\%$) and red corresponds to 17 applications of IMP (density to $1.44\%$).

To the best of our knowledge, this has not been previously identified and provides new insight into why IMP has found success in discovering winning tickets (Frankle & Carbin, 2019; Frankle et al., 2020a), as well as has found general success in the study of DNNs (Frankle et al., 2020b).

## 3 RESULTS

### 3.1 UNIVERSALITY IN THE IMP FLOW

We start by examining the IMP flow of computer vision models. The goal of this analysis is to examine DNNs that are known to allow for the successful transfer of winning tickets, and determine whether they have the same relevant and irrelevant parameters. If they do, this would confirm a major prediction of the theory developed in Sec. 2.

For spin systems in statistical physics, the RG flow is studied in the space of coupling constants (Fig. 1). Each coupling constant is usually assumed to be the same for all spins, greatly reducing the dimensionality of the underlying space. While most DNNs do not set all parameters of a given type to the same value, we can none-the-less estimate the "influence" the parameters of a given layer or residual block have on the full DNN, by considering the functions

$$M_i(n) = \sum_{j=1}^{N^{(i)}} m_j^{(i)}(n) \Big/ \sum_{k=1}^{N} m_k(n), \tag{13}$$

where $M_i(n)$ describes the percentage of all non-zero parameters that are in layer or residual block $i$ after IMP iteration $n$. The numerator sums over all $N^{(i)}$ of the elements of the pruning mask restricted to layer or residual block $i$, $m^{(i)}$. Each $m_j^{(i)} \in \{0, 1\}$. The denominator sums the entire pruning mask, making it equivalent to the number of non-pruned parameters in the DNN. Given the analogous role the $M_i$ play to the coupling constants of spin systems, we expect that they may be eigenfunctions of $\mathcal{I}$, and, therefore, may encode the relevant and irrelevant components of a given DNN.

Because ResNet-50 has four residual blocks, IMP induces a four-dimensional flow in the space of $M_i$. To visualize this trajectory, we examined two-dimensional slices. Examples of these are plotted in Fig. 2 for ResNet-50, pre-trained on ImageNet via simCLR self-supervised training (Chen et al., 2020b). The flow for three different tasks are shown: ImageNet, CIFAR-10, and CIFAR-100. This data comes from experiments performed by Chen et al. (2021a).

If the $M_i$ are eigenfunctions of the IMP operator, then they will scale exponentially with respect to the number of iterations $\mathcal{I}$ has been applied. That is, $M_i(n + 1) = \mathcal{I}M_i(n) = \lambda_i M_i(n) = \lambda_i^{n+1} M(0)$, where $\lambda_i$ is the eigenvalue governing its exponential growth/decay. We can find $\lambda_i$ by the simple inversion, $\lambda_i = M_i(n + 1)/M_i(n)$. As long as $M_i(n) \neq 0$, this is well defined

Table 2: Computed eigenvalues of the eigenfunctions $M_i$ for ResNet-50, pre-trained on ImageNet using simCLR (Chen et al., 2020b). Red (green) corresponds to those eigenvalues that are relevant (irrelevant). Any eigenvalues within a standard deviation of $\lambda = 1$ are considered marginal and colored cyan.

| Task | $\lambda_1$ | $\lambda_2$ | $\lambda_3$ | $\lambda_4$ |
|---|---|---|---|---|
| ImageNet | $1.15 \pm 0.01$ | $1.09 \pm 0.02$ | $1.02 \pm 0.01$ | $0.95 \pm 0.02$ |
| CIFAR-10 | $1.11 \pm 0.01$ | $1.06 \pm 0.01$ | $1.01 \pm 0.01$ | $0.98 \pm 0.002$ |
| CIFAR-100 | $1.08 \pm 0.03$ | $1.04 \pm 0.02$ | $1.00 \pm 0.01$ | $0.99 \pm 0.01$ |

and we can compute $\lambda_i$ by doing this for each pair of data points. Performing this on our dataset yields estimates of eigenvalues with standard deviations $< 5\%$ (Table 2), suggesting that it is indeed reasonable to consider the $M_i$ as being eigenfunctions of $\mathcal{I}$. Additionally, we find that ImageNet, CIFAR-10, and CIFAR-100 have similar distributions in eigenvalues (Table 2). In particular, all three identify the first two residual blocks as being relevant and identify the last block as being marginal or irrelevant. From the theory developed in Sec. 2, these results imply that all three models are in similar universality classes and that winning tickets can be transferred between them. Chen et al. (2021a) indeed confirmed this, finding that winning tickets found on ImageNet could be successfully transferred to both CIFAR-10 and CIFAR-100.

This universality in IMP eigenvalues may be due to the fact that the first few residual blocks learn low level statistics of the data (Neyshabur et al., 2020), which scale in specific, non-trivial ways for natural images (Ruderman & Bialek, 1994; Saremi & Sejnowski, 2013). The sensitivity to the first two residual blocks, which are the furthest from the output layer, is in contrast to standard spin systems. In the case of the usual two-dimensional Ising model, the RG removes long-range interactions (i.e. couplings between next-nearest neighbors, next next-nearest neighbors, etc.), which are typically assumed, for physical reasons, to be weak. Systems with relevant long-range interactions have been found to have unique properties, such as the existence of a phase transition in one-dimension (Dyson, 1969), which is known not to happen for the standard Ising model. The importance of these long-range interactions may therefore be another interpretation for the benefit of pre-training and normalized initialization (e.g. He initialization) for convolutional DNNs, as they are presumably part of what sets the ResNet-50 to be in this "long-range interaction" regime.

However, we emphasize that the universal IMP eigenvalues (Table 2) are not merely a trivial reflection of the pre-training and normalized initialization. In particular, the specific task matters, as each model starts from a slightly different point in the four-dimensional space of $M_i$ [ImageNet: (0.0079, 0.0557, 0.3055, 0.6309); CIFAR-10: (0.0075, 0.0556, 0.3176, 0.6193); CIFAR-100: (0.0070, 0.0519, 0.3004, 0.6407)], implying that each task arrives at distinct distributions of parameters, despite starting from the same initial set of parameters. While pre-training and normalized initialization likely constrain the degree of these differences, it is conceivable that there are tasks, which would sufficiently re-organize the parameter distributions, such that earlier blocks could become irrelevant and/or later blocks could become relevant. In such cases, RG theory predicts that these tasks would not admit the transfer of winning tickets. Future work should attempt to identify such tasks.

Similar results as those reported in Table 2 were found for ResNet-50, pre-trained on ImageNet using supervised learning (Huh et al., 2016) (Apppendix A), which was also previously shown to have successful winning ticket transfer between tasks (Chen et al., 2021a). While the theory developed in Sec. 2 made no assumption on architecture, we also computed IMP eigenvalues on large scale lottery ticket experiments using VGG-16 and BERT models, finding again that the $M_i$ could reasonably be considered eigenfunctions of $\mathcal{I}$ (Appendix B). This further supports our claim that the RG framework is general and can be broadly used to study winning ticket transferability.

Table 3: Computed eigenvalues of the eigenfunctions $M_i$ for the first stage of various ResNet architectures. Same coloring as in Table 2.

| ResNet architecture | Stage 1 |
|---|---|
| ResNet-14 | 1.07 |
| ResNet-20 | 1.07, 1.07 |
| ResNet-32 | 0.99, 1.06, 1.06, 1.08 |
| ResNet-44 | 1.00, 0.98, 1.06, 1.06, 1.05, 1.07 |
| ResNet-56 | 1.02, 0.98, 1.07, 1.06, 1.05, 1.06, 1.04, 1.08 |

## 3.2 ELASTIC LOTTERY TICKET HYPOTHESIS

Recent work on the "Elastic Lottery Ticket Hypothesis" (E-LTH) has extended the notion of tranferability by finding it possible to transform winning tickets found on one DNN, to another with a different architecture (Chen et al., 2021c). In particular, it was found that, for architectures in the same family (e.g. ResNets) trained on the same task (e.g. CIFAR-10), it was possible to either squeeze (by removing residual blocks) or stretch (by replicating residual blocks) winning tickets so that they could be transferred.

Three results from this study of the E-LTH are especially noteworthy:

1. The smallest ResNet considered (ResNet-14) had the weakest transferability properties;

2. The more unique residual blocks replicated, the better the performance;

3. Removing the later residual blocks, in the case of shrinking, or replicating the earlier residual blocks, in the case of stretching, led to the best results.

While preliminary hypotheses on the origin of these results were developed by Chen et al. (2021c), involving dynamical systems and unrolled estimation, detailed theoretical understanding was left to future work. Given that the RG framework developed above led to insight into the nature of the transfer of winning tickets in the case of different tasks, we wondered whether it could similarly help to explain the nature of transfer in the case of different architectures.

As discussed in Sec. 2, two tasks that have the same IMP eigenvalues can transfer winning tickets. Therefore, we expect that transforming a given winning ticket from one architecture (source) to another (target) will be successful if its IMP flow matches that of the target. In particular, the success (or lack-thereof) should rely on whether the number of relevant, irrelevant, and marginal residual blocks in the target architecture is matched by the transformed source ticket.

To test this, we computed the eigenvalues associated with the eigenfunctions $M_i$ (Eq. 13), for each "normal" residual block of ResNet-14, ResNet-20, ResNet-32, ResNet-44, and ResNet-56, trained on CIFAR-10 [data from Chen et al. (2021c)]. As discussed in the original paper, the downsampling residual block in each stage was not transformed between winning tickets, so they were not considered. The eigenvalues for the residual blocks in the first stage are presented in Table 3.

We find that, while the smaller ResNets (ResNet-14 and ResNet-20) have only relevant residual blocks, the larger ResNets have at least one non-relevant block. This offers a straightforward explanation as to why ResNet-14 had particularly poor transferability: using it as the source ticket necessarily leads to the transformed ticket having only relevant residual blocks, which does not match the target tickets' structure. Having only relevant residual blocks may make the first stage too sensitive as the depth of the ResNet is increased, leading to poorer performance. This is likely related to the idea that smaller ResNet models are not sufficiently expressive, but has the benefit of being theoretically backed and quantitative. Relatedly, we find that replicating few unique residual blocks (i.e. replicating only block 1 or blocks 1 and 2) can lead to an over representation of either relevant or non-relevant blocks. This mismatch is predicted by the theory to lead to worse transferability.

Finally, we find that most of the non-relevant residual blocks are in the early part of the first stage. Therefore, removing the first several blocks when shrinking a winning ticket (e.g. dropping the first four blocks to go from ResNet-56 to ResNet-32), or replicating the last several blocks when

stretching a winning ticket (e.g. adding the last two blocks to go from ResNet-32 to ResNet-44), will lead to too many relevant residual blocks. Again, this will make the transformed source ticket have a different structure than the target ticket and thus, will lead to worse performance.

Note that for the later two stages, the nature of the distribution of eigenvalues is different (Tables 7 and 8 of Appendix C). In particular, stage 2 has only relevant and marginal blocks, and stage 3 has only marginal and non-relevant blocks. However, there is consistent evidence for the fact that the number of each type of block is related to the three results from the original E-LTH paper (Chen et al., 2021c) we highlighted.

## 4 DISCUSSION

Inspired by similarities between the current state of sparse machine learning and the state of statistical physics in the early-to-mid–20$^{th}$ century, we proved that iterative magnitude pruning (IMP), the principle method used to discover winning tickets, is a renormalization group (RG) scheme. Given that the development of the RG led to a first principled understanding of universal behavior near phase transitions, as well as a way in which to characterize materials by such behavior, we reasoned that viewing IMP from an RG perspective may lead to new insight on the universality of winning tickets (Desai et al., 2019; Mehta, 2019; Morcos et al., 2019; Soelen & Sheppard, 2019; Chen et al., 2020a; Gohil et al., 2020; Chen et al., 2021a; Sabatelli et al., 2021) and the general success IMP has found in the study of DNNs (Frankle et al., 2020b).

By viewing IMP as inducing a flow in DNN parameter space (Fig. 2), we showed that ResNet-50, pre-trained on ImageNet, has trajectories with similar properties for different tasks (Table 2). As winning tickets are known to be transferable between these tasks Chen et al. (2021a), these results support a key prediction of the theory developed in Sec. 2. Applying the RG framework to recent work which extended the notion of winning ticket universality by transforming tickets between different architectures ("Elastic Lottery Ticket Hypothesis"), we again found that the IMP eigenvalues allowed for a new, quantitative perspective on experimental results (Table 3). While RG theory may be unfamiliar, and simpler explanations to the phenomena may conceivably exist, to our knowledge, no current theory has provided as general and successful a framework for understanding winning ticket universality, as what we have present here.

A wealth of theory has been developed around the RG, including a number of numerical and theoretical tools that go beyond what we have used. Given the connection between IMP and the RG made explicit in this work, we believe that there is considerable potential for collaboration between the two fields. Possible directions of future study include:

- Bringing the RG framework to other sparsificiation methods, which may potentially similarly be viewed as RG schemes;
- Bringing the RG framework to study winning tickets beyond computer vision models, such as in natural language processing (Chen et al., 2020a; Prasanna et al., 2020; Yu et al., 2020), reinforcement learning (Yu et al., 2020), and lifelong learning (Chen et al., 2021b);
- Computing and classifying systems by their critical exponents (such as $\gamma$ in Eq. 1). We attempted to do this, but we were limited by having only a few independent seeds and a scaling function with multiple free parameters (Rosenfeld et al., 2020). This led to large differences between computed critical exponents of systems that displayed qualitatively similar scaling behavior (see Appendix D). Using more advanced methods, such as finite scaling (Goldenfeld, 2005), and increasing the number of independent seeds may alleviate this problem;
- Exploring the connection between the RG framework developed here and work done to build an effective theory of DNN behavior (Roberts et al., 2021).

Finally, while RG theory provides a way to know whether two given tasks, or architectures, can have winning tickets transferred between them, knowing what the minimal density a transferable winning ticket can have, remains an open question. In order to address this, finite size effects (Goldenfeld, 2005), differences in symmetries, and non-linear corrections to the IMP flow (Raju et al., 2019) may need to be taken into account. This interesting, and important, question will be the focus of future work.

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

## A   ADDITIONAL RESNET-50 TRANSFER RESULTS

In the main text, we computed IMP eigenvalues for ResNet-50, pre-trained on ImageNet using simCLR self-supervised training (Chen et al., 2020b). These models were shown to have the best transferability of winning tickets across tasks in Chen et al. (2021a). However, other, pre-trained models were also found to have good transferability. Here, we present the same analysis for ResNet-50, pre-trained on ImageNet using supervised training (Huh et al., 2016), again finding that the winning tickets for each task have similar distributions of eigenvalues (Table 4). The data used for this was also taken from Chen et al. (2021a).

Table 4: Computed eigenvalues of the eigenfunctions, $M_i$, for ResNet-50, pre-trained on ImageNet using supervised learning (Huh et al., 2016). Red (green) corresponds to those eigenvalues that are relevant (irrelevant). Any eigenvalues within a standard deviation of $\lambda = 1$ are considered marginal and colored cyan.

| Task | $\lambda_1$ | $\lambda_2$ | $\lambda_3$ | $\lambda_4$ |
|---|---|---|---|---|
| ImageNet | $1.13 \pm 0.01$ | $1.08 \pm 0.01$ | $1.02 \pm 0.01$ | $0.96 \pm 0.01$ |
| CIFAR-10 | $1.04 \pm 0.04$ | $1.06 \pm 0.03$ | $1.03 \pm 0.03$ | $0.96 \pm 0.02$ |
| CIFAR-100 | $1.04 \pm 0.05$ | $1.06 \pm 0.02$ | $1.03 \pm 0.004$ | $0.97 \pm 0.01$ |

## B   ADDITIONAL EVIDENCE OF IMP EIGENFUNCTIONS IN VGG-16 AND BERT

In the main text, we presented results for ResNets. However, there is no explicit assumptions of architecture present in the theory developed in Sec. 2. Therefore, we explored whether the functions $M_i$ (Eq. 13) were similarly eigenfunctions for VGG-16, trained on CIFAR-10, and BERT, trained on the self-supervised mask language modeling (MLM) task. Data is from experiments performed by Zhang et al. (2021) and Chen et al. (2020a), respectively.

As can be seen in Tables 5 and 6, for both pairs of architectures and tasks, we find that the computed eigenvalues have small (usually $< 5\%$) standard deviations, suggesting that it is reasonable to consider them as eigenfunctions of $\mathcal{I}$. Note that we only considered the first 17 applications of IMP (matching that of the ResNet experiments presented in Table 2), in the case of VGG-16. Considering additional applications led to higher standard deviations, which is expected as the size of the system decreases and finite size effects start to become non-negligible.

Interestingly, while VGG has layers that are relevant, marginal, and irrelevant, like ResNet-50, BERT has only marginal layers. This suggests that BERT may have been pushed into a fixed point of the IMP flow. Whether, and how, this is related to the challenges associated with going deeper with Transformers, will be studied of future work.

Table 5: Computed eigenvalues of the eigenfunctions, $M_i$, for VGG-16 trained on CIFAR-10. Note that only the prunable layers had their IMP eigenvalues computed. Coloring is same as Table 4.

| | |
|---|---|
| $\lambda_1$ | $1.22 \pm 0.02$ |
| $\lambda_4$ | $1.16 \pm 0.02$ |
| $\lambda_8$ | $1.15 \pm 0.04$ |
| $\lambda_{11}$ | $1.14 \pm 0.04$ |
| $\lambda_{15}$ | $1.12 \pm 0.05$ |
| $\lambda_{18}$ | $1.09 \pm 0.06$ |
| $\lambda_{21}$ | $1.07 \pm 0.07$ |
| $\lambda_{25}$ | $1.03 \pm 0.07$ |
| $\lambda_{28}$ | $0.95 \pm 0.05$ |
| $\lambda_{31}$ | $0.91 \pm 0.03$ |
| $\lambda_{35}$ | $0.89 \pm 0.03$ |
| $\lambda_{38}$ | $0.88 \pm 0.04$ |
| $\lambda_{41}$ | $0.87 \pm 0.05$ |

Table 6: Computed eigenvalues of the eigenfunctions, $M_i$, for BERT trained on the MLM task. Coloring is same as Table 4.

| | |
|---|---|
| $\lambda_1$ | $0.99 \pm 0.01$ |
| $\lambda_2$ | $1.00 \pm 0.01$ |
| $\lambda_3$ | $1.00 \pm 0.01$ |
| $\lambda_4$ | $1.00 \pm 0.01$ |
| $\lambda_5$ | $1.01 \pm 0.01$ |
| $\lambda_6$ | $1.01 \pm 0.01$ |
| $\lambda_7$ | $1.01 \pm 0.01$ |
| $\lambda_8$ | $1.00 \pm 0.01$ |
| $\lambda_9$ | $1.00 \pm 0.002$ |
| $\lambda_{10}$ | $1.00 \pm 0.001$ |
| $\lambda_{11}$ | $1.00 \pm 0.01$ |
| $\lambda_{12}$ | $1.00 \pm 0.01$ |

## C  ADDITIONAL E-LTH RESULTS

In the main text on the, the experimental results surrounding the "Elastic Lottery Ticket Hypothesis" focused only on the eigenvalues corresponding to the eigenfunctions of residual blocks in the first stage. The models considered in Chen et al. (2021c) had two other stages, whose eigenvalues we report here in Tables 7 and 8.

While the distribution of eigenvalues is different than it is in stage 1 (i.e. considerably fewer relevant residual blocks), they again support the idea that the distributions are related to the results from Chen et al. (2021c) that we focused on in the main text.

Table 7: Computed eigenvalues of the eigenfunctions $M_i$ for the second stage of various ResNet architectures. Coloring is same as Table 4.

| ResNet architecture | Stage 2 |
|---------------------|---------|
| ResNet-14 | 1.03 |
| ResNet-20 | 1.02, 1.02 |
| ResNet-32 | 1.02, 1.03, 1.01, 1.00 |
| ResNet-44 | 1.04, 1.03, 1.03, 1.03, 1.00, 1.01 |
| ResNet-56 | 1.03, 1.02, 1.02, 1.02, 1.01, 1.00, 0.99, 1.00 |

Table 8: Computed eigenvalues of the eigenfunctions $M_i$ for the third stage of various ResNet architectures. Coloring is same as Table 4.

| ResNet architecture | Stage 3 |
|---------------------|---------|
| ResNet-14 | 0.95 |
| ResNet-20 | 0.97, 0.95 |
| ResNet-32 | 0.99, 0.99, 0.96, 0.95 |
| ResNet-44 | 0.99, 0.98, 0.97, 0.95, 0.96, 0.96 |
| ResNet-56 | 0.99, 0.98, 0.98, 0.97, 0.96, 0.97, 0.98, 0.96 |

## D  CRITICAL EXPONENTS

Following the recent work on scaling laws for IMP developed by Rosenfeld et al. (2020), we fit the error, $\epsilon$, defined as $100\%$ minus the top-1 accuracy, as a function of density, $d$, defined as percent of weights remaining, via the functional form

$$
\begin{aligned}
\hat{\epsilon}(d, \epsilon_{np}, \epsilon^{\uparrow}, \gamma, p) &= \epsilon_{np} \left\| \frac{d - ip\left(\frac{\epsilon^{\uparrow}}{\epsilon_{np}}\right)^{\frac{1}{\gamma}}}{d - ip} \right\|^{\gamma} \\
&= \epsilon_{np} \left[ \frac{\left(d^2 + p^2(\frac{\epsilon^{\uparrow}}{\epsilon_{np}})^{2/\gamma}\right)}{(d^2 + p^2)} \right]^{\gamma/2},
\end{aligned}
\tag{14}
$$

where $i = \sqrt{-1}$. Here $\epsilon_{np}$ is interpreted as the error associated with not pruning, $\epsilon^{\uparrow}$ as the asymptotic error upon maximal pruning, $\gamma$ as the sensitivity of the combination of network architecture, task, activation function, and optimizer to pruning, and $p$ as controlling how the transition from no change in error to power-law scaling takes place. Importantly, $\gamma$ can be viewed as a critical exponent under the RG framework. Because systems in the same universality class have the same critical exponents, we wondered if we could find evidence of universality in the computed value of $\gamma$ for various different computer vision models.

We found the fits to be sensitive to $\epsilon_{np}$ and, in some cases, there was not a clear single value for $\epsilon_{np}$. Therefore, we included $\epsilon_{np}$ as a free parameter (which Rosenfeld et al. (2020) did not do),

Table 9: Critical exponent, $\gamma$, for ResNet-50 with either no pre-training and 5% rewinding, or pre-training using ImageNet, simCLR, or MoCo. Bottom half gives $\gamma$ computed on the pre-trained task after pre-training the parameters. Data from Chen et al. (2021a).

| Data-set | No pre-train | ImageNet | simCLR | MoCo |
|----------|--------------|----------|--------|------|
| CIFAR-10 | $\gamma = -0.14$ | $\gamma = -0.10$ | $\gamma = -0.16$ | $\gamma = -0.21$ |
| CIFAR-100 | $\gamma = -0.11$ | $\gamma = -0.15$ | $\gamma = -0.06$ | $\gamma = -0.24$ |
| SVHN | $\gamma = -0.07$ | $\gamma = -0.06$ | $\gamma = -0.03$ | $\gamma = -0.29$ |
| ImageNet | – | $\gamma = -0.42$ | – | – |
| simCLR | – | – | $\gamma = -1.01$ | – |

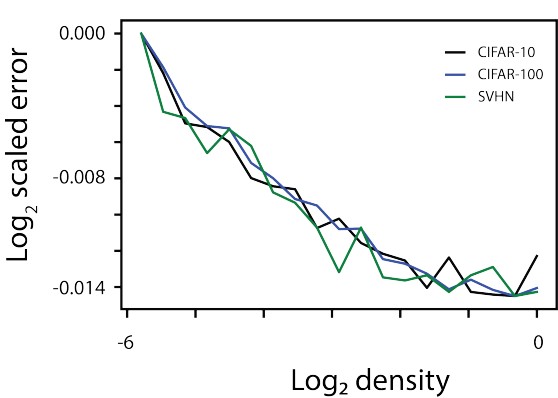

Figure 3: **Error as a function of density scaling of CIFAR-10/100 and SVHN.** Overlaying the error as a function of density curves for ResNet-50, trained from random initialization, on CIFAR-10/100 or SVHN, with 5% rewind. The overlaid curves shows very similar, but noisy, behavior.

but we tightly bounded the value so as to minimize instability in adding another free parameter. To numerically compute the fits, we used scipy's curve fitting function: scipy.optimize.curve_fit (Jones et al., 2001–).

The computed values of $\gamma$ for ResNet-50 evaluated on CIFAR-10, CIFAR-100, and SVHN are presented in Table 9. The effect of pre-training using various state-of-the-art methods is also given. The $\gamma$ values vary, sometimes considerably, across tasks and method of pre-training (e.g. CIFAR-10 and SVHN pre-trained via simCLR). However, we note that the data came from only two or three seeds, making the fits susceptible to noise and making the quantification of error in the computed $\gamma$ difficult. In addition, the fitting function of Eq. 14 has multiple free parameters (including an additional one we added), again making the fit susceptible to noise.

Taking these possible complications into account, we overlaid the different error as a function of density scaling curves (an example of which is given in Fig 3). We found that they qualitatively matched each other more than the computed $\gamma$ may have made it seem. We imagine that trying different fitting procedures (especially making use of methods developed in the context of statistical physics, such as finite scaling (Goldenfeld, 2005)), as well as using more independent random seeds, will enable more accurate and robust approximations of $\gamma$.

