# OpenReview forum: "Universality of Deep Neural Network Lottery Tickets: A Renormalization Group Perspective"
_ICLR.cc/2022/Conference — ICLR 2022 Submitted_

### Official Review · Reviewer_i8Gz · 2021-11-01

**Correctness:** 2
**Technical Novelty And Significance:** 2
**Empirical Novelty And Significance:** 1
**Recommendation:** 3
**Confidence:** 3

**Main Review:**

### Strength
 - The paper is easy to follow.
- The motivation of trying to explain the transferrable of winning tickets using the renormalization group seems interesting.
- The paper attempts to build a theoretical basis for understanding the iterative magnitude pruning, which is widely used in works related to the lottery ticket hypothesis.

### Weakness
Authors claim the paper provides experimental support of the renormalization group theory in large scale lottery ticket experiments. However, I have some questions about the experiments.
- For the experiments on the Elastic Lottery Ticket Hypothesis, the datasets used to train different networks (Table 3) seems not introduced in the paper. Do we still have the same conclusion for the networks trained on different datasets, e.g., CIFAR10 and ImageNet?
- The authors provide some experimental results/conclusions and try to explain them using the renormalization group. However, this explanation doesn't answer the question proposed at the beginning of the work - "whether a given ticket can be transferred to a given task." I wonder if authors could provide experiments in a way that the theory helps us find a winning ticket that can be transferred to some tasks where previous studies cannot find. Such experiments can help readers understand how to use the theory in practice for searching a winning ticket for a specific task.

Another strong claim in the paper is that the authors mention that the work provides new tools for classifying combinations of DNN architectures, activation functions, and optimizers. Similar to the previous question, I don't quite understand how to leverage the introduced theory of renormalization group in practice to achieve that.

**Summary Of The Paper:**

The work aims to provide a deeper understanding of a corollary in the lottery ticket hypothesis: how to know whether a winning ticket found in one task can be transferred to another task. The paper tries to combine the widely used pruning method in the lottery ticket hypothesis (iterative magnitude pruning) with a concept from statistical physics, which is the renormalization group. The authors show iterative magnitude pruning can be considered as a renormalization group operator. Experiments are conducted using the renormalization group to examine whether pruned models are in the same universality classes and extend the renormalization group to the Elastic Lottery Ticket Hypothesis.

**Summary Of The Review:**

While trying to obtain a prior for which tasks of a given winning ticket can be transferrable using the theory from statistical physics seems interesting, the paper doesn't provide enough experimental results to show how to use such an explanation to improve iterative magnitude pruning or determine the best architecture that can be transferred for different tasks. The work is more like working in the progress report, and more results can help strengthen the work.

---

> ### Author Response · Authors · 2021-11-19
> **Response to Reviewer i8Gz**
>
> A. Thank you for pointing this out. All Elastic Lottery Ticket Hypothesis (E-LTH) experiments were done with ResNets trained on CIFAR-10. This has been added to Sec. 3.2.
>
> B. As discussed in the general response (above), we agree that the phrasing of this claim was confusing and suggested something that was not supported by our experiments. Hopefully, with the changed language and the additional experiments in Sec. 3.1, this will be more clear and better reflected by the experiments we performed.
>
> C. Similarly to the point in B, we agree that this was not well explained in the original manuscript. The point we were trying to make was that, since the RG theory is general, if the IMP eigenvalues of two models (i.e. specific combinations of architectures, activation functions, optimizers, and tasks) are computed, then by looking at whether their relevant and irrelevant residual blocks match up, we can predict whether winning tickets can be transferred between them (like the analysis that is now performed in Sec. 3.1 of the revised manuscript). We have changed the wording of this claim, and again hope that the changed language and the additional experiments in Sec. 3.1 will make this clearer.

---

> ### Author Response · Authors · 2021-11-26
> **Response to Rebuttal**
>
> Dear Reviewer i8Gz,
>
> We were wondering whether our rebuttal addressed your concerns? If you have any additional comments or questions, please let us know so we can attempt to answer them. Thank you!

---

### Official Review · Reviewer_iuDr · 2021-11-02

**Correctness:** 2
**Technical Novelty And Significance:** 3
**Empirical Novelty And Significance:** 3
**Recommendation:** 5
**Confidence:** 3

**Main Review:**

The paper is well written. The ideas of bringing renormalization group theory to explain the lottery ticket hypothesis theory is very interesting. However, I have some confusions need to be clarified.

1). The analogy between the IMP and the Ising model, i.e. equation (1) and (2) is a bit far-fetched to me. The two equations appear alike, however, I donot quite see the intuition behind it. For example, one can argue that the magnetization (order) disappeared when the temperate increases in spin system, however, the error (disorder) on the other hand decreased when the density increases in DNN, almost two opposite directions.

2). In the section 2.2 where the RG and IMP is proved connected, specifically in equation 12, I donot see how the amplitude of the parameters plays in a role in defining the RG operator. One of the key factors in IMP is that the selection of parameters is based on the amplitude. If the iterative pruning is done randomly without considering the amplitude, is the RG frame still work? In addition, the operator (equation 12) seems only consider per layer operation, but the IMP often performed per model. Could the author clarify the difference and how that could impact the operator?

3). Another main concern is the IMP flow results on ResNet50 (Fig. 2). The normalized initializations scale the initial weights according to the channel dimensions, the consequence of which is smaller weight amplitude on the later ResBlocks. During the IMP, the percentage of pruning elements in each ResBlock will naturally follow the amplitude ratio. Since the eigenfunction of the IMP operator is defined based on the pruning rate in each ResBlock, the result IMP flow in Figure 2 should just follow the amplitude difference between blocks. Not sure how this result could have supported the theory. Besides, equation 13 does not include amplitude term as well.


**Summary Of The Paper:**

This paper tries to find a theoretical explanation of the transferability of lottery ticket used in similar tasks. Observing the similarity between the universality in renormalization group and the lottery ticket hypothesis, the author proposes that the iterative magnitude pruning, which is used to find the winning tickets, could be a renormalization group scheme. The authors also provide some evidence on their theory on vision model of ResNet families.

**Summary Of The Review:**

In general, I think this paper is well written and proposed an interesting theory, however, the argument and proof are not very convincing to me.

---

> ### Author Response · Authors · 2021-11-19
> **Response to Reviewer iuDr**
>
> 1) We agree that the phrasing of this could be misleading and suggest that we were implying a correspondence between magnetization and error. The observation we were trying to highlight was that, for both spin systems and DNNs, important quantities exhibit power-law scaling in a non-trivial regime. As it is known that critical phenomena in statistical physics can be identified and categorized by this scaling, we had hoped to use this similarity to further motivate the application of RG theory to sparse DNNs. The wording of this has been changed in the revised manuscript to emphasize this more accurately.
>
> 2) As discussed in the general response (above), we believe this question of parameter magnitude being missing in the proof that the IMP is an RG scheme comes from the fact that it was not clear that the analogous quantity to spins is unit activations, not DNN parameters (Table 1). While the parameters are the ones getting removed, the effect of this on the activations is a coarse-graining, where they become more independent of each other and less affected by inputs coming from previous layers. To make this point more clear, we have amended Fig. 1 and explicitly mentioned this point in the derivation in Sec. 2.2. Also note that the masking operator, which in the case of IMP performs pruning based on parameter magnitude, is present in Eq. 12. This makes the projection operator dependent on parameter magnitude.
>
> 3) The question of the role of normalized initialization in the exponential scaling of $M_i$ is a good one and we agree that this was not sufficiently addressed in our initial manuscript. We have now added more discussion and experiments to Sec. 3.1 to make it clear that: 1) our results come from ResNet-50s that has been trained; 2) different tasks induce different distributions of parameter magnitudes across residual blocks, even when starting from the same initial model. We believe that, while normalized initialization likely plays an important role in constraining the degree to which the distributions can differ, it is not trivially causing the exponential scaling of $M_i$ we observe. We have added additional discussion around this in Sec. 3.1.

---

> ### Author Response · Authors · 2021-11-26
> **Response to Rebuttal**
>
> Dear Reviewer iuDr,
>
> We were wondering whether our rebuttal addressed your concerns? If you have any additional comments or questions, please let us know so we can attempt to answer them. Thank you!

---

### Official Review · Reviewer_cSyR · 2021-11-03

**Correctness:** 3
**Technical Novelty And Significance:** 3
**Empirical Novelty And Significance:** 3
**Recommendation:** 5
**Confidence:** 5

**Main Review:**

Strengths:
The paper tries to tackle an interesting and important question on the understanding of LTH ticket properties. I also appreciate that the paper focused on trying to explain experiments from prior work and utilized realistic experiments that are of interest to machine learning practitioners.

Weaknesses:
While I appreciate the high-level conceptual connection between RG and IMP, I am not fully convinced that the connection is really a precise one, or that it provides strong quantitative "explainability" for LTH phenomenon (that is, RG can be used to explain things that a simpler hypothesis can not). On the other hand, I appreciate that there is some preliminary evidence for this in the paper. Perhaps there are two primary ways in which the paper can be strengthened:
(1) Either a more extensive set of experiments (also of other predictions from RG) that would demonstrate that RG is really a good explanation for the phenomenon.
(2) Ruling out some simpler hypotheses, that don't invoke RG, that can explain the phenomenon.

Stated otherwise, I don't find the experimental support thus far to be a precision "test" that RG is a useful theory for describing the observations. For example:
--Exponential scaling of the M_i(t). Is there something about constraints from IMP and dynamics of weights during gradient descent that would roughly lead to exponential scaling for such a macroscopic / coarse-grained variable?
--"Universality" in Table 2. This experiment suggests that where the bulk of the pruning is done (which blocks) is set by the architecture / topology. Perhaps a more "precise" test of RG here would be if it could be used to tell us *why* topology largely categorizes the universality class?
--I'm not sure how Figure 2 shows universality, except for the cases where there is scaling collapse.
--In Sec. 3.2, in discussion on how the smallest ResNet has the weakest transferability: a simpler explanation could be that a smaller model is not expressive enough to learn a good representation useful for more expressive architectures. Maybe this is correlated with aspects of RG, but my point is that there is a simpler mechanism behind it.

Overall, I think this paper has potential and has preliminary evidence in a promising direction, but I think more support would be necessary to convince a reader that the full machinery of RG is crucial for explainability of LTH phenomenon.

Question: pg. 8, paragraph 4. Where it says "...which does not match the source tickets' structure", should it read "target" tickets'?


**Summary Of The Paper:**

This paper seeks to explain empirical observations in the literature on the Lottery Ticket Hypothesis (LTH) based on ideas from renormalization group (RG) theory in physics. The authors focus on the particular case of iterative magnitude pruning (IMP) as the pruning scheme and view the flow on the space of model parameters during IMP as analogous to RG flow in the space of couplings of a Hamiltonian. The primary experimental evidence for the connection comes from two considerations. (1) Computing the percentage of all non-zero parameters that are in a particular residual block (of a ResNet model) at a particular time. For ResNet-50 models considered, this leads to four scalar, dynamical quantities. These four quantities are found to grow or shrink exponentially with time, suggesting they are eigenfunctions of the IMP map. For three different types of learning & dataset (ImageNet / simCLR / MoCo) and one architecture (ResNet-50), the authors find the same eigenvalue across the three settings, per block. From this, the authors conclude that this universality (shared exponent) across different settings is evidence for the analogy to RG. (2) Transferability of lottery tickets. The authors hypothesize that the relevance / irrelevance / marginality  of a particular block ought to be matched by the source and target architecture when transferring tickets. More specifically, there are three conclusions from the literature (#1-3 in Sec 3.2) that are explained (for example, why the smallest model has the weakest transferability).

**Summary Of The Review:**

This paper takes a bold step in connecting a theoretical framework from physics to a set of important empirical observations in machine learning (LTH). It presents preliminary evidence that this connection could be useful, but I think more support is needed to show that the machinery of RG is crucial and predictive.

---

> ### Author Response · Authors · 2021-11-19
> **Response to Reviewer cSyR**
>
> As discussed in the general response (above), we have tried to add additional clarification as to why, from the standard RG approach in statistical physics, it is reasonable to study $M_i$ and expect that they might be eigenfunctions of the IMP operator. In particular, the $M_i$ are similar to the coupling constants in spin systems (as they measure the “influence” the parameters of each residual block exert on the rest of the DNN). In addition, we find that different tasks induce different topological arrangements, as the ResNet-50 we study in Sec. 3.1 starts from a different point in the four-dimensional space of $M_i$ depending on which specific task it is applied to. This suggests that the $M_i$ are affected by the task, and yet still have the same relevant and irrelevant residual blocks. Given that it was known that winning tickets could be transferred between the tasks we studied, this was seen as supporting the theory’s prediction that IMP eigenvalues are related to transferability.
>
> We appreciate the fact that the full machinery of the RG may seem more complicated than what is necessary for explaining individual phenomena related to the Lottery Ticket Hypothesis. We certainly agree that simpler explanations may exist. However, as noted in the general response (above), we believe that our work shows that: 1) because the IMP is an RG scheme, RG theory is an appropriate language with which to discuss IMP; 2) the RG framework provides quantitative results for the universality of winning tickets across both tasks and architectures, which no other existing theory has shown; 3) the large body of literature surrounding RG theory suggests that there are many other theoretical and numerical tools that the field of sparse machine learning can draw from, allowing future work to go beyond what we have studied here. Discussion of all of this has been added to Secs. 3.2 and 4.
>
> Thank you for pointing out this typo. We have fixed it in the revised manuscript.

---

> ### Author Response · Authors · 2021-11-26
> **Response to Rebuttal**
>
> Dear Reviewer cSyR,
>
> We were wondering whether our rebuttal addressed your concerns? If you have any additional comments or questions, please let us know so we can attempt to answer them. Thank you!

---

### Official Review · Reviewer_ud5a · 2021-11-03

**Correctness:** 3
**Technical Novelty And Significance:** 3
**Empirical Novelty And Significance:** 3
**Recommendation:** 6
**Confidence:** 3

**Main Review:**

Strength:

+ This paper has a very clear motivation and tries to answer a very important question about the Lottery Ticket Hypothesis.

+ It provides a theoretical basis for understanding the success of IMP and the universality of winning tickets.

+ To show the universality in the IMP flow, extensive experiments are conducted based on various important methods in different settings: supervised ImageNet training, and self-supervised simCLR and MoCo training.

+ The insights and finding from this paper have potential to inspire a future explorations between IMP and RG.


Concerns and questions:

- Authors claim that RG theory can provide a way to predict which combinations of task, optimizer, activation function, and architecture can have winning tickets transferred between them. Is there any experimental evidences related to pruning to support such claims?

- The experiments and analysis are mainly about RestNet based backbone. How about other commonly used backbone, e.g. VGG, DenseNet, etc?

**Summary Of The Paper:**

Although Lottery Ticket Hypothesis has suggested an exciting corollary about the winning tickets, it is still unclear why winning ticket universality exists, or any way of knowing a priori. In this paper, authors make use of renormalization group theory to perform detailed understanding of this hypothesis. Authors find that the principle method iterative magnitude pruning (IMP) is directly related to the renormalization group (RG).

Based on the contributions of  RG leading to a first principled understanding of the universality in behavior near phase transitions, viewing the IMP from an RG perspective may lead to new insight on the universality of winning tickets. Such insight is build upon the experimental support of the theory in large scale lottery ticket experiments.

**Summary Of The Review:**

Overall, the paper has a good quality with a clear motivation, it tries to answer a very important question about the Lottery Ticket Hypothesis. Authors' findings about the relationships between principle method iterative magnitude pruning and the renormalization group may lead to new insight on the universality of winning tickets. Such insight is important and can inspire a considerable potential for future collaboration between IMP and RG.

---

> ### Author Response · Authors · 2021-11-19
> **Response to Reviewer ud5a**
>
> A. As mentioned in the general response (above), we agree that the phrasing of this claim was confusing and suggested something that was not supported by our experiments. Hopefully, with the changed language and the additional experiments in Sec. 3.1, our claim, and the fact that it is supported by our experiments, will be more clear.
>
> B. As mentioned in the general response (above), there is no theoretical reason to believe that the RG framework will not be able to apply to DNNs based on other backbones. However, we appreciate that, since this perspective is new, directly supporting this would make our case stronger. We therefore have added experimental results on VGG-16 and BERT models (new Appendix B). In both cases, we show that we can find IMP eigenvalues using the same approach that we used for the ResNet models (Eq. 13). Future work should dive deeper into these different architectures and what their different distributions of eigenvalues imply.

---

> ### Author Response · Authors · 2021-11-26
> **Response to Rebuttal**
>
> Dear Reviewer ud5a,
>
> We were wondering whether our rebuttal addressed your concerns? If you have any additional comments or questions, please let us know so we can attempt to answer them. Thank you!

---

### Author Response · Authors · 2021-11-19
**General response to reviewers (part 1)**

We thank the reviewers for their time and feedback.

Your comments made it clear that several major points of the manuscript were not sufficiently communicated and/or were not well supported. We have therefore strived to improve the manuscript by ensuring clearer discussion and better experimental support of these points. These changes, which we believe significantly increase the quality and rigor of the manuscript, are discussed below:

A. The proof that iterative magnitude pruning (IMP) is a renormalization group (RG) scheme (Eq. 12) relies heavily on the fact that the spins in the RG picture are analogous to the unit activations in the deep neural network (DNN) picture. While this was stated in Table 1, the schematic in Fig. 1 was misleading and made it natural to assume that the analogous quantity was instead the DNN parameters. Therefore, we have edited Fig. 1 to be more accurate and have explicitly referenced the analogous quantities in Sec. 2.2.

B. The interpretation of the universality present in the ResNet experiments we performed rested on the fact that the $M_i$ were eigenfunctions of the IMP operator. As was mentioned by several reviewers, the choice of $M_i$ seemed arbitrary and the cause of its exponential scaling could arise from reasons like normalized initialization. In statistical physics, the eigenfunctions of the RG operator, which define the relevant and irrelevant directions, are combinations of the coupling constants (see Fig. 1, RG flow). As noted in Table 1, the coupling constants are analogous to the DNN parameters. While spin systems usually assume that all spins have the same value of each coupling type, DNNs do not have all parameters of the same type take the same value. However, we could still estimate how much “influence” the parameters of each residual block has on the rest of the system, by considering what proportion of non-pruned parameters remained in that residual block (i.e. $M_i$). Because the $M_i$ play a similar role to the coupling constants, we therefore expected them to be eigenfunctions (which, was indeed supported by experiments). This explanation has been added to the text of Sec. 3.1 to make the choice of $M_i$ more transparent, and more grounded in the RG theory.

As for the cause of the exponential scaling of M_i, we note that the IMP eigenvalues (which control this scaling) were computed for a DNN that had already been trained from initialization. Therefore, while we agree that normalized initialization likely played an important role in determining the distribution of parameter magnitudes (as we had mentioned in Sec. 3.1), we do not believe it is solely for this reason there was a similar distribution of eigenvalues. Indeed, we now find support for different tasks leading to different starting values of each M_i (for more on these new results, see point D below). We add more discussion on this in Sec. 3.1.

---

> ### Author Response · Authors · 2021-11-19
> **General response to reviewers (part 2)**
>
> C. Several of the reviewers pointed out that there may be other (simpler) ways in which to understand the winning ticket universality. We whole-heartedly agree. However, as we mention in Sec. 2.1, existing conceptual frameworks are non-trivial to apply in practice. Indeed, to the best of our knowledge, there has been no general, quantitative framework that has the broad experimental support our work does.
>
> To make these points more clear, we have added additional discussion in Secs. 3.2 and 4. We particularly emphasize that, while other theories that explain the transferability of winning tickets may exist, the RG framework developed in our work is an appropriate theory (because IMP is an RG scheme) that yields results. Additionally, as we discuss in Sec. 4, RG theory has decades worth of development, and future synthesis between the two fields may lead to further advances.
>
> D. The choice of wording in the statement that RG theory could “a priori say which combinations of tasks, optimizer, activation function, and architecture would allow for transfer of winning tickets” made it sound like it could allow for predicting the success of transfer experiments without doing any pruning. As several reviewers correctly pointed out, there was no experimental support of this. The point we had been trying to make was that, if in the future there existed a database of IMP eigenvalues for various DNN models, then researchers could know which of those previously characterized models a new model could be successfully transferred to. However, this can only happen once that new model’s IMP eigenvalues are computed. Note that this kind of classification by IMP eigenvalues is analogous to what the field of statistical physics does by classifying materials by their RG eigenvalues.
>
> The language of this point has been changed to more accurately make the intended point. Additionally, we realized that the experiments we chose to analyze were not the most optimal ones to illustrate this. We have, therefore, analyzed additional experiments and find that tasks that have been shown to allow for the successful transfer of winning tickets (e.g. CIFAR-10, CIFAR-100, and ImageNet) do indeed have the same relevant and irrelevant residual blocks (Table 1 and Table 5 in Appendix A of the revised manuscript). These results support a key prediction of our theory, and additionally show how future experiments could be analyzed and the success of transferability be predicted.
>
> E. Because we only studied experiments involving the ResNet architectures, one reviewer questioned the generalizability of the theory to other, common families of DNNs. We note that, since the proof of IMP being a RG scheme does not rely on any assumptions of architecture, there is no reason to believe that our framework will be limited. However, to support this, we add a new appendix, Appendix B, where we show that we can find IMP eigenvalues for BERT and VGG-16, using the same approach (Eq. 13). This is additional evidence that the developed theory is broadly applicable to studying winning ticket universality.
>
> ---------
> For more details on the above revisions, and for our responses to other additional comments, please see the responses to each reviewer below and our revised manuscript.
>
> Lastly, please note that while several major changes were made to address the highlighted weaknesses, the underlying theory and arguments have remained the same.

---

### Decision · Program_Chairs · 2022-01-20

**Decision:**

Reject

**Comment:**

This paper observes the similarity between the universality in renormalization group and the lottery ticket hypothesis and proposes that the iterative magnitude pruning, which is used to find the winning tickets, could be a renormalization group scheme. The authors also provide some evidence on their theory on vision model of ResNet families. While it is interesting to try a theoretical explanation of the transferability of lottery ticket used in similar tasks using the theory from statistical physics, the paper does not provide enough experimental results to show how to use such an explanation to improve iterative magnitude pruning or determine the best architecture that can be transferred for different tasks. Therefore the work is more like working in the progress report and not ready for publication yet.